# RB2: Robotic Manipulation Benchmarking with a Twist

**Sudeep Dasari**[1], **Jianren Wang**[1], **Joyce Hong**[1], **Shikhar Bahl**[1], **Yixin Lin**[2], **Austin Wang**[2],
**Abitha Thankaraj**[3], **Karanbir Chahal**[3], **Berk Calli**[4], **Saurabh Gupta**[5], **David Held**[1],
**Lerrel Pinto**[3], **Deepak Pathak**[1], **Vikash Kumar**[2], **Abhinav Gupta**[1,2]

**CMU**[1], **FAIR**[2], **NYU**[3], **WPI**[4], **UIUC**[5]

https://www.rb2.info

## Abstract

Benchmarks offer a scientific way to compare algorithms using objective performance metrics. Good benchmarks have two features: (a) they should be widely useful for many research groups; (b) and they should produce reproducible findings. In robotic manipulation research, there is a trade-off between reproducibility and broad accessibility. If the benchmark is kept restrictive (fixed hardware, objects), the numbers are reproducible but the setup becomes less general. On the other hand, a benchmark could be a loose set of protocols (e.g. object set [9]) but the underlying variation in setups make the results non-reproducible. In this paper, we re-imagine benchmarking for robotic manipulation as state-of-the-art algorithmic implementations, *alongside* the usual set of tasks and experimental protocols. The added baseline implementations will provide a way to easily recreate SOTA numbers in a new local robotic setup, thus providing credible *relative rankings* between existing approaches and new work. However, these "local rankings" could vary between different setups. To resolve this issue, we build a mechanism for pooling experimental data between labs, and thus we establish a single global ranking for existing (and proposed) SOTA algorithms. Our benchmark, called Ranking-Based Robotics Benchmark (RB2), is evaluated on tasks that are inspired from clinically validated Southampton Hand Assessment Procedures [27]. Our benchmark was run across two different labs and reveals several surprising findings. For example, extremely simple baselines like open-loop behavior cloning, outperform more complicated models (e.g. closed loop, RNN, Offline-RL, etc.) that are preferred by the field. We hope our fellow researchers will use RB2 to improve their research's quality and rigor.

## 1 Introduction

Imagine a new roboticist tasked with building an ice cream scooping robot. Which existing algorithm could best accomplish this task? Even seasoned roboticists would find it hard to answer this question, because most published methods claim improved performance over baselines. Unfortunately, many of these claims are mirages, constructed by incomparable setups, subjective task definitions, and/or implementation mistakes [21]. This unscientific approach makes it hard for the field to discern Gold from Pyrite. In practice, this introduces a rich gets richer bias – big and established labs can focus on optimizing methods for their *own* setup while comparing against their *own* past work. In contrast, new entrants find it needlessly hard to reproduce simple experiments, let alone push the state of the art. It is clear that the status quo needs to change.

35th Conference on Neural Information Processing Systems (NeurIPS 2021) Track on Datasets and Benchmarks.

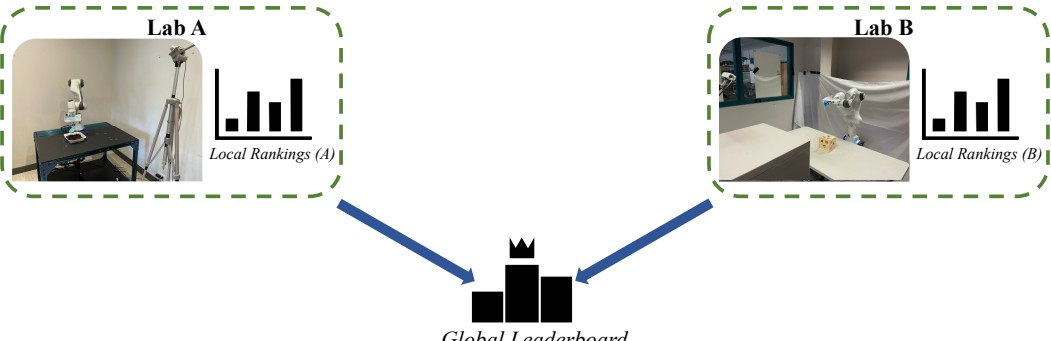

Figure 1: We present RB2, a real-world robot learning benchmark consisting of four manipulation tasks needed in daily human activity: pouring, scooping, zipping, and insertion. We provide experimentation, training and evaluation procedures as well as implementations of several state-of-the-art robot learning methods. Code, documentation and other details can be found at: https://www.rb2.info .

The obvious solution to this problem is benchmarking. Benchmarks offer an easy way to compare algorithms using scientific performance metrics. These objective performance scores allow researchers to fairly evaluate new algorithms, and enable the field as a whole to judge which methods "actually work." Furthermore, benchmarks meaningfully reduce the barrier to entry, since newcomers can focus on their algorithm and evaluating it on the benchmark – the experimental setup is fixed and the baseline performance on benchmark is available as well. A good benchmark must appeal to a **wide audience** (i.e. it should cover many groups' use cases) while also being easily **reproducible**. In many machine learning adjacent fields – like Natural Language Processing and Computer Vision – these goals were more easily achieved. It is now standard practice to distribute data across the world, and test metrics can be easily reproduced by evaluating models on the same held out test set.

However, in robotics, benchmarking comes with a fundamental trade-off between broad accessibility and reproducibility. Performance on real robotic hardware cannot be accurately modelled in simulation, especially in tasks requiring rich gripper-object contacts. This necessitates *physical* experimental setups, that are difficult to exactly replicate in new lab settings. One option would be to create a restrictive benchmark by pre-deciding all environmental variables (e.g. a single well defined task, fixed hardware choices, specified experimental protocol) so that performance numbers are comparable across papers. However, this directly results in a less general benchmark since different labs might have different hardware/constraints. On the other end of spectrum, a benchmark could just contain basic experimental setup descriptions. For example, YCB objects [9] only standardizes object sets. However, this makes experiments impossible to reproduce, since changes in the setup (e.g. different action rate) could advantage some methods over others.

In this paper, we rethink the concept of benchmarking in the context of real-world robotic manipulation. Our key insight is that building precisely reproducible robotic setups is impossible and therefore absolute performance numbers on a benchmark task are meaningless. For example, we find that running the same pouring experiment in two different lab spaces, changes our performance metrics by 20%. However, if the performance numbers for both the baselines and the new proposed algorithm are re-computed at same physical location and under the same setup, they are likely to be comparable and hence produce meaningful relative ranking. We note that a truly superior approach would outperform all baselines for majority of different setups. Therefore, we propose to build a **manipulation benchmark that helps users establish local relative ranking between different methods and uses local rankings from several researchers to develop global rankings across different robotics setups**. This is achieved by treating a robotic manipulation benchmark as a set of experimental protocols, *and* a model zoo with state-of-the-art baselines. This will provide a way to quickly recreate baseline numbers in a new robotic setup, and thus provide credible baselines and comparisons to any researcher.

We introduce a new robotics benchmark RB2 (Ranking Based Robotics Benchmark). Our benchmark tasks are inspired by the Southampton Hand Assessment Procedure (SHAP) [27], a standard test for assessing dexterity in occupational therapy via daily-living manipulation tasks. RB2 consists of four such tasks: pouring, scooping, zipping, and insertion. All of these tasks can be successfully performed with a standard 2-Finger gripper and commodity robot hardware – like the Franka Pandas. For each task, we list all the test and train objects, and carefully document objective performance metrics

like the amount of material successfully poured. The key feature of our benchmark is the multi-task setup. Our focus is on ranking general-purpose manipulation algorithms and not individual tasks which can be engineered to produce high numbers. We plan to run multiple tracks of training/testing frameworks including but not limited to: models trained on static offline datasets, pre-trained models that are finetuned with active learning (e.g. Reinforcement Learning), and active models trained from scratch. We will a maintain model zoo and protocols for each of these tracks. For the initial version, we focus on the offline track and provide 5 relevant baseline algorithms. This allows other researchers – even those interested in better hardware or fancier learning algorithms – to easily contextualize their gains against strong initial baselines. We demonstrate the credibility of our benchmark philosophy and framework by evaluating same set of algorithms across two different labs.

## 2 Related Work

**Robotics Benchmarks**   There is a lot of inherent uncertainty when designing real world experiments, and small variations in environmental factors can have disproportionate effects on the performances of different methods [49]. There are many sources of variation in robotic experiments ranging from different hardware, to lighting conditions, to the evaluation tasks themselves. One common solution is to control for all of these other factors, by building fully replicate-able environments [29; 47]. Many groups have attempted to address these problems via benchmarks. For example, standardized object sets like YCB [9] seek to control for variance due to object selection, as most algorithms are highly sensitive to the test object set. However, objects are only part of the experimental procedure. Benchmarks in robotics exist for specific tasks such as picking and placing [33], grasping [5; 6], bin picking [11; 30; 32], include less common applications such as aerial [42] and cloth manipulation [20]. Some are tailored to specific hardware, such as robot hands [1; 13], tri-finger manipulators [19], bi-manual manipulation [10], humanoids [25] or grasping gripper design [16; 36]. However, these are often too rigid in their hardware and experimentation requirements. As a result, such controlled setups tend to target smaller research communities. Insisting on such precise control often excludes researchers who lack the necessary hardware, or are interested in more general tasks. For example, a roboticist cannot easily re-tool a highly controlled pick-place environment in order to perform household scooping experiments. Low-cost robotics platforms [35; 47] seek to democratize access to these setups, but often have limited hardware/control capabilities and are thus constrained to simple manipulation tasks.

**Simulation Benchmarks**   Simulation often provides a lower-cost alternative to real-world experiments. While benchmarking has struggled to gain traction in the real world, it is common place in simulation. In particular, reinforcement learning algorithms are often tested in simulated robotic tasks suites [43; 7; 48; 50; 23; 28]. Offline reinforcement learning benchmarks [18; 31] provide various datasets obtained from agents deployed in simulation. Furthermore, benchmarks like SAPIEN [46] and ManipulaThor [15] allow researchers to test algorithm's ability to learn semantic concepts. However, simulators have difficulty in reproducing the visual diversity of the real world thus only allow for learning relatively simple semantic concepts. Moreover, these simulators often rely on physics engines [45; 12] that cannot model the nuances of real world dynamics, complex object interactions, or hand-object contacts. The simplifications made by these simulators often allow the algorithm to "game" the task, and thus it is difficult to draw inferences about the effectiveness of the particular method, especially in the wild.

## 3 Benchmarking with a Twist

A benchmark is typically defined as a set of tasks each with precisely defined evaluation functions. Each evaluation function $\rho$ numerically scores a given method $\phi_j$ on a target task $\tau_i$ using a pre-defined protocol (parameterized by $k_j, \theta_i, m_i$ respectively). Thus, we can define each measure as $\rho_i(\phi_j(k_j), \tau_i(\theta_i), m_i)$. For benchmarks to fairly and reproducibly evaluate methods, all other variables ($\theta_i, m_i$) should be fixed in advance. As a result, each evaluation becomes a pure function of the given method: $\hat{\rho}_i(\phi_j(k_j))$. The resulting benchmark $\hat{B}(\phi_j(k_j)) = \{\hat{\rho}_i(\phi_j(k_j))\}_{i=0}^n$ enables researchers to independently and robustly evaluate their proposed contributions.

However, robotics environment parameters $\theta_i$ depend on real-world factors and therefore can not be tightly controlled. For example, robotic hardware, lab condition (e.g. lighting, work-space size, etc.), and underlying physical parameters (e.g. friction) could all change from one lab to another. This

makes it impossible to ship a single benchmark $\hat{B}$ across labs. So, how can we overcome this barrier and design a better benchmarking scheme for robotics?

Lets take a cue from how roboticists handle similar situation in a simulated setting. Consider an ice-cream scooping task with evaluation function defined as $\rho_i(\phi_j, \tau_i(\theta_c), m_i)$, where $\theta_c$ denotes relevant system parameters (e.g. ice cream's temperature and scoop's friction). Let's assume a prior State-Of-The-Art (SOTA) baseline $\phi_A(k_A)$ is publicly available and reports the following performance on the task: $\rho_A = \rho_i(\phi_A(k_A), \tau_i(\theta_c), m_i)$. However, a researcher believes she has developed a better algorithm - $\phi_B$ - for warm conditions where ice cream melts and makes the handle slippery (i.e. $\theta_w$). Thus, she creates a new evaluation $- \rho_w(\phi, \tau_i(\theta_w), m_i)$, and evaluates both methods. In other words, she runs experiments in order to calculate $\rho_w^{(A)}$ and $\rho_w^{(B)}$. If $\rho_w^{(B)} > \rho_w^{(A)}$, the researcher can safely conclude her new method is indeed better in the warmer setting! Note that she does not directly compare against $\rho_A$ since system parameters changed, but she did use it to find *relevant baselines* to compare against. After all, her analysis would be less valuable if $\phi_A$ was a weak baseline (instead of SOTA).

Our benchmark RB2 builds on this insight. We redefine robotics benchmarks as a joint set of tasks (w/ evaluation function) and prior SOTA baselines. When a new researcher wants to test his/her algorithm, they first create a new local evaluation function which uses same tasks (e.g., pouring) but with different local environment parameters. The researcher then evaluates their proposed work *alongside* the given baselines on the target tasks. While the raw results are valuable for analysis, they will likely differ wildly between labs. Thus, we propose rigorously ranking baselines using a fixed function $R$. The final benchmark creates a set of rankings that measure how the proposed work stacks up against the baselines. These local rankings should be more reproducible across environments given suitable choice of ranking function $R$. If a new algorithm significantly outperforms the current SOTA, the code for this algorithm could be contributed as a new baseline.

But do these rankings remain consistent for different $\theta_t$? That is, if different labs run same set of methods will they report similar rankings? We argue that obtaining perfect rank orderings at a local site (though individual results are still valid) might be impossible due to differences in environment parameters. However, a truly better method would consistently perform other baseline methods for majority of $\theta_i$s. Inspired by this observation, our benchmark proposes to develop global rankings by using several local rankings at different labs. Note that individual users will not need to run experiments in multiple labs, but will instead upload their results to a central repository. As a result, the *field* will distribute the work required to compute a global ranking. We hope that this process of aggregating data from multiple experiments in various labs over time, will allow us to infer stronger claims about global rankings as a community.

## 4 Benchmark Details - Tasks and Experimental Protocols

**Setup:** Operationally, environments must provide access to observations $o_t$, which typically consists of robot sensors and cameras. They also need to accept actions $a_t$ that control the agent (e.g. actuator commands, target joint positions, etc.). Actions are applied for a fixed period of time $\delta t$ before the next observation is generated $o_{t+1}$. A sequence of these observations and actions form a trajectory $\mathcal{T} = \{o_0, a_0, o_1, a_2, \ldots, o_H\}$, where $H$ is a horizon (i.e. maximum length). We require that all robot testing environments follow this "recipe".

**Tasks ($\tau$):** First, we note that "task" is an overloaded term in robotics, and thus we must be careful to avoid confusion. We define a *task* to be a broad category of behaviors (e.g. pouring), while an *instance* is akin to a sample – e.g. "scoop almonds from the black box on the right." Each task must provide success measures that objectively determine how effectively a given trajectory solved a particular instance.

**Task/Environment Parameters ($\theta$):** Since users will separately build environments, system parameters will naturally vary between setups. We decompose task parameters $\theta_i$ into workspace parameters $\theta_i^w$ and system parameters $\theta_i^s$. Workspace parameters constitute task-specification parameters, such as initial / goal object positions etc, which must be tightly controlled by the user. In contrast, system parameters are (often) outside of users control. These include robot hardware, precise sensor characteristics, lighting conditions, etc. In our benchmarks, we provide concrete definitions for the workspace parameters. On the other hand, system parameters are free to be determined by the experimenter, except for three constraints: (1) Only one morphological embodiment can be utilized, i.e. the robot cannot be changed manually between experiments (automatic tool changing is allowed).

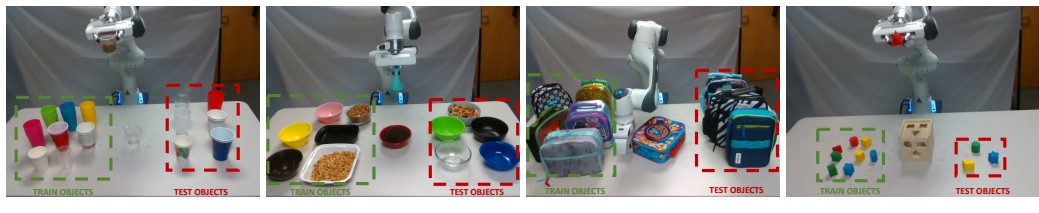

| (a) Pouring | (b) Scooping | (c) Zipping | (d) Insertion |

Figure 2: We present four manipulation tasks: pouring, scooping, zipping and insertion as part of RB2. Each task involves a set of train (green) and (red) test objects.

(2) No sensor or actuation augmentation within the workspace is allowed (all sensors and actuators must be outside the task area at the start of the experiment). (3) A robot's initial location should be outside the workspace (the defined task area). Examples of properly constructed lab spaces are available in Fig. 1.

**Baselines ($\phi$):** Under our philosophy, SOTA baselines $\phi_i$ are now integral parts of the overall benchmark. A baseline (i.e. policy) should produce actions given an observation ($a_t \sim \pi(a_t|o_t)$) that result in an successful trajectory for a given task instance.

**Local Ranking ($R$):** Recall that relevant baseline performance $\rho_i(\phi_j(k_j), \tau(\theta_i^w, \theta_i^s), m_i)$ must be reevaluated in each lab environment, due to shifting environmental factors. Once evaluation (for baselines and new method) is complete, the data is processed into local rankings – e.g. ordering of methods by performance based on experiments. While prior evaluation benchmarks simply compare mean task performance to form rankings, we adopt a more rigorous approach and test for statistical significance. This is done to reduce noise in rankings. Specifically, we employ Tukey's HSD test [34] which performs pairwise t-tests between means (of task metrics $\rho$) $\mu_1, ..., \mu_M$, using the Studentised Range Distribution, which has statistic, $q = (\mu_{\max} - \mu_{\min}) * \sqrt{\frac{n}{2S^2}}$, where $n$ is the pooled sample size and $S$ is the pooled standard deviation.

**Global Ranking**: Instead of purely relying on local rankings, which could be noisy, performance data from multiple labs is aggregated in a central dataset. Global rankings are then calculated for each baseline, using the Plackett-Luce [38] method for combining a set of rankings into a global ranking. For a given set of local rankings $\{L_1, L_2, ..., L_N\}$, which could either contain the full set of baselines or a partial set, Plackett-Luce produces a distribution over the methods, using a score $s_i$ called the "worth". The method then uses Maximum Likelihood Estimation to solve for the global ranking, according to the distribution of $s_i$.

For the sake of explanation, consider the following concrete example. Say that RB2 has 5 baseline algorithms implementations ($B_1$-$B_5$). Currently the global rankings show: $B_3 > B_2 > B_5 > B_4 > B_1$. A new research group proposes a new algorithm $A_1$ after our baselines are pushed. To demonstrate their algorithm is better they download the code of the top-3 baselines ($B_3$, $B_2$ and $B_5$) and compare their proposed method ($A_1$) to the baselines (a typical use-case). This comparison is published using local rankings. In order to aggregate these results into the global ranking, the authors submit their paper, results and code to our repository. To build a global ranking, we consider their local ranking orderings ($A_1, B_3, B_2$, etc). Note that this ranking can be partial (only have $m$ out of $N$ possible methods) or have possible ties in the rankings. We employ the Plackett-Luce method to aggregate these into a global ranking, using the procedure described previously. Once the code is submitted the top-3 suggested baselines now include $A_1$.

However, note that the global rankings still does not include $A_1$, since the experiment was not verified by an independent research group. If another group proposes method $A_2$, they will have to download code for $A_1, B_3$ and $B_2$ (the top 3 current baselines) and compare against these. Similarly to $A_1$, they will run experiments, collect local rankings and results which they will submit to our repository alongside their implementation. Once $A_2$ uploads their local rankings, our global rankings get updated to reflect $A_1$ as the new leader, since their results have been confirmed by another lab. Just like before, $A_2$ will be a added as a suggested baseline, but will need further verifcation to be on the global leaderboard.

# 5 Benchmark Instantiation: RB2

On a high level RB2 consists of three stages repeated per task: (1) demonstration data is collected by a human operating in training environments; (2) the dataset is used to fit a learned policy using a baseline algorithm; (3) the policy's control performance is evaluated in various test scenarios. We now outline; the exact experimental procedures, the tasks considered, and the implemented baselines.

## 5.1 Collection and Evaluation Protocols ($m$):

**Dataset Creation:** In principle, one could randomly sample train instances by placing random objects from the office in front of a robot. In practice, this would create potential blind-spots and result in unrepeatable findings. To fix this failure mode, we resort to stratified sampling. Specifically, each task uses $O_{tr}$ train objects and $P_{tr}$ train positions randomly sampled from the work-space, resulting in $N = O_{tr} \times P_{tr}$ training instances. Expert demonstrators collect an optimal trajectory for each train task instance, resulting in a final dataset of training trajectories $\mathcal{D} = \{T_1, \ldots, T_N\}$.

**Testing Procedure:** After training, each policy is evaluated on a distribution of test instances, which consists of $P_t$ test positions (not in train set) and a combination of $O_t$ test objects and the original $O_{tr}$ train objects. The evaluation proceeds by executing policies in each test instance, and recording task-specific performance metrics (e.g. amount of material poured) based on the trajectories they generate. Instead of letting a human operative subjectively determine success, it is calculated by combining and thresholding in a consistent fashion (per task). This scheme allows us to perform repeatable experiments, while minimizing systemic bias due to human inconsistency. Furthermore, it enables rigorous testing of how policies adjust to unseen spatial configurations and novel objects. Note that train positions are explicitly not contained in these test instances, to prevent rewarding policies that are severely over-fit.

## 5.2 Tasks and Workspace Parameters ($\tau, \theta^w$)

This benchmarking protocol is now applied to each considered task. Specifically, RB2 consists of 4 household manipulation tasks that humans encounter on a daily basis; pouring, scooping, insertion, and zipping. These tasks are loosely inspired by the SHAP tests [27] and were chosen to be represent a broad range of human manipulation. Specific per-task design decisions are now presented in detail. For object lists, please refer to the benchmark website.

**Pouring:** This task consists of taking 50 grams of material – in our case almonds – to a single cup placed on the table, and pouring all the material without spilling. Note that this is a single stage task, so the robot starts the trajectory with a cup containing the material in hand. Performance is measured by the amount of material successfully poured into the target cup, and is normalized by the amount of material originally in the cup (i.e. $\frac{\text{poured}}{\text{total}}$). $O_{tr} = 9$ train cups and $O_t = 6$ test cups are drawn from a broad distribution of durable household cups. $P_{tr} = 15$ training positions and $P_t = 8$ testing positions are randomly sampled from the work-space, resulting in $N = 135$ training instances and $T = 136$ testing instances. Both object sets and a prototypical task are illustrated in Fig. 2a.

**Scooping:** The robot is initialized with a spoon in its gripper and presented with a bowl filled with material. The robot must successfully control the spoon to scoop material from the bowl without spilling. The operator measures the amount scooped and the maximum amount of material the spoon could possibly hold. Performance is measured by $\frac{\text{amount scooped}}{\text{max amount}}$. Note in this case a training/testing "object" consists of a bowl-material pair. We consider $O_{tr} = 21$ training objects in this tasks (7 bowls, 3 materials), and $O_t = 5$ testing objects (5 bowls, 1 material). Training instances use $P_{tr} = 6$ positions, while testing instances have $P_t = 5$ positions. Objects and setup are visualized in Fig. 2b.

**Zipping:** The episode begins with a lunchbox placed in the workspace, with its zipper grasped by the robot's gripper. The robot must then successfully unzip the bag without loosing hold of the zipper. Note that a heavy weight is placed in the bag to prevent it from frequently falling out of the workspace. The operator measures the distance zipped by the robot as a fraction of the bag's total zip distance (i.e. $\frac{d_{zip}}{d_{bag}}$). A trial is considered successful if the fraction zipped is $\geq 0.5$. This task uses $O_{tr} = 6$ training lunchboxes, and $O_t = 3$ testing lunchboxes. $P_{tr} = 10$ train positions are generated by randomly rotating and shifting the bag, while being careful to not remove the zipper from the robot gripper. $P_t = 5$ test positions are generated using a similar procedure. The objects used and a prototypical task instance are shown in Fig. 2c.

**Insertion:** In this task, a common children's insertion toy is placed in front of the robot, and a block is placed in its gripper. The robot must successfully insert the block into the matching hole. Success is reported using a binary metric: 1 if the block is inserted into the hole and 0 otherwise. We use $O_{tr} = 9$ training blocks (i.e. objects) corresponding to different holes on the toy, and $O_t = 3$ test blocks from an unseen (during training) face on the toy. We sample $P_{tr} = 7$ different train positions and $P_t = 5$ test positions by randomly rotating the block. This results in a total of $N = 63$ train instances and $T = 60$ test instances. The objects and tasks are visualized in Fig. 2d.

## 5.3 Baselines

**Baseline System Parameters:** The requirements defined in Section 3 lead us to adopt a simple (and common) joint position control stack for baseline algorithms that runs at 30 Hz. Actions are specified as joint targets, which are translated into motor control signals using an underlying high-frequency PD controller. Observations consist of RGB images and the robot's proprioceptive signals (e.g. angles from joint encoders). Pictures of our setup can be found in Fig. 2.

We will release the baseline system parameters and stacks for the Franka Panda so that other researchers can use this stack for a rapid prototyping of baseline algorithms in their environment. We also hope several researchers will release their algorithms and stacks so that followup work can replicate the results in their environment.

## 5.4 Network Architecture and Representation Learning

Note that any visuo-motor policy must be able to accurately localize objects in the scene. To help solve this problem we provide a vision stack for pre-training representations. Specifically, a simple neural network (3 VGG convolutions followed by Spatial Softmax [17]) is trained to predict object poses ($p$) in robot coordinate space from an image ($i$) of the object placed in front of the robot. Given a dataset of 1200 $(i, p)$ pairs collected per environment, a network $F$ is trained to minimize $||F(i) - p||_1$. For stable performance, we ensure object positions sufficiently cover the workspace, and that the robot hand position and gripper (e.g. place objects in fingers) are randomized. Once trained, representations from the network $R(i)$ are used to initialize the learned policies.

Note that other groups need not tie their methods to this specific vision pipeline. Indeed, we encourage researchers to develop better robotic vision stacks, and believe that RB2 could offer a (so-far missing) way to scientifically compare vision stacks for manipulation research. However, learning general 3D object representations using weak supervision (e.g. demonstration trajectories) is far from a solved problem. Thus, we provided this representation learning pipeline as a way for manipulation/control focused researchers to sidestep this challenging vision problem.

## 5.5 Open-Loop Imitation

Open-loop behavior cloning is a simple yet effective method for learning robotic control strategies from demonstration data-sets. Given an initial observation from a trajectory, the policy should predict $H$ actions taken by the expert at *every* time-step. Those predicted actions are then executed on the robot without re-planning. More specifically, the expert actions are concatenated into a single vector $a = [a_0, \ldots, a_{H-1}]^T$, and the policy is optimized using the following supervised loss: $\min_\pi ||\pi(o_0) - a||_1$. Note that the policy – $\pi(o_0) = G(R(o_0))$ – is initialized with pre-trained representations and then fine-tuned. During test time, $\pi$ predicts an action trajectory given the initial observation, which is then executed on the robot.

**Smoother Control via NDPs:** In addition to this traditional behavior cloning objective, RB2 provides an additional implementation of Neural Dynamic Policies (NDPs) [3; 4]. This baseline works as a smoother version of open-loop behavior cloning, by embedding dynamical system structure (described by DMPs [41; 39; 14]) inside the learned network. In practice, NDPs process an input state (i.e. image of the scene) and outputs parameters for a dynamical system (DMP), which include the goal and the weights for the forcing function for the system. The network integrates the dynamical system and outputs a trajectory for the robot to follow. NDPs are able to reason in the space of physically smooth trajectories and thus are able to output more meaningful behavior as compared to open-loop behavior cloning.

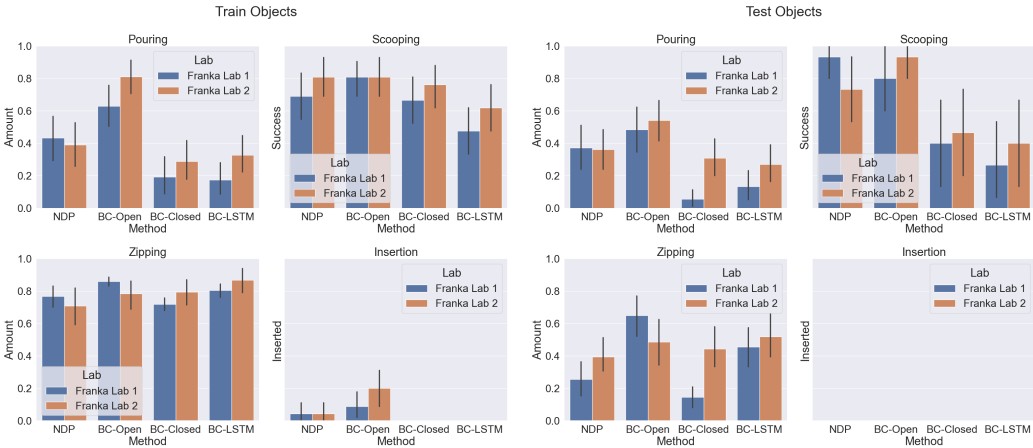

Figure 3: Results of five baselines on four tasks. By definition, metrics are normalized to [0,1], w/ 1 being best possible performance. As the results show Open-Loop BC often outperforms NDP, and closed loop baselines. Note Offline RL is omitted since it never succeeds for any task.

## 5.6 Closed-Loop Imitation

A common failure mode in open-loop policy learning occurs when the policy makes a mistake: since action trajectories are not re-planned at every time-step the learned controllers cannot adjust to correct for errors made earlier. Closed loop behavior cloning fixes this failure mode, by querying the policy for new actions at every time-step. Expert trajectories are broken into observation-action tuples $(o_t, a_t)$, and the policy is trained to predict the next action given current observation. This results in the following objective: $\min_\pi ||\pi(o_t) - a_t||_1$. The closed loop policy is bootstrapped from pre-trained representations, just like in the open-loop case. During testing, the current observation is fed into the policy and the predicted action is executed on the robot.

**Recurrent Networks:** Prior work [40] demonstrated that Recurrent neural networks can improve behavior cloning performance in situations where the expert strategy is not perfectly Markovian (e.g. pauses before scooping, etc.). Thus, Our final baseline pairs closed loop behavior cloning with a LSTM [22]. Specifically, we train LSTMs to mimic a length $T$ trajectory "snippet" $\{o_i, a_i, o_{i+1}, \ldots, o_{i+T}, a_{i+T}\}$ via teacher forcing. The supervised loss, initialization scheme, and robot deployment strategy remain unchanged from the prior closed-loop behavior cloning baseline.

## 5.7 Offline Reinforcement Learning

While reinforcement learning (RL) has shown promise in robot learning [37; 29; 2] training a policy to perform manipulation tasks in the real world is challenging and time consuming. Offline RL, addresses this problem by training policies on static datasets [26; 24; 18]. To this end, we provide an implementation of MOReL [24]. MOReL firstly learn a dynamics model $f(s_t, a_t) = s_{t+1}$ over the data. The dataset collection scheme provided by RB2 consists of optimal trajectories $\tau$, noisy trajectories $\tau + \epsilon_i$, where $\epsilon_i$ are different levels of Gaussian noise randomly sampled, as well as random interaction data. The demonstrations are manually labeled and determines when task success is achieved in the trajectory (say at timestep $k$). The reward $r(a_t, s_t)$ is 0 for $t < k$ and 1 for $t \geq k$.

MOReL then builds a pessimistic MDP by dividing the task region into known and unknown, called the Unknown State-Action Detector (USAD). This allows adding a large negative reward to unknown regions of the task space, to avoid large distribution shifts. In particular, this is done via using an ensemble of models $f_1, f_2, ..., f_M$ and considering the ensemble discrepancy = $\max_{i,j} ||f_i(a_t, s_t) - f_j(a_t, s_t)||$. After the USAD is computed, MOReL uses an MPC [44] planner initialized with a policy trained via behavior cloning.

## 6 Experiments

Recall that RB2 consists of 4 tasks, 5 baselines, and a central reporting repository. Our experiments seek to verify each of the components. Specifically, each baseline is evaluated on every task in *multiple* lab environments. Data produces from these experiments are analyzed to answer the following questions: (1) are RB2's baselines correctly implemented and if so what do they reveal about current SOTA approaches? (2) Can statistically significant rankings be determined on a local

| Method | BC-Open | NDP | BC-Closed | BC-LSTM | MOReL |
|---|---|---|---|---|---|
| *Pouring*: | | | | | |
| Franka Lab 1 | 1 | 2 | 2 | 3 | 5 |
| Franka Lab 2 | 1 | 2 | 3 | 3 | 5 |
| **Global** | 1 | 2 | 3 | 3 | 5 |
| *Scooping*: | | | | | |
| Franka Lab 1 | 1 | 2 | 3 | 3 | 5 |
| Franka Lab 2 | 1 | 1 | 3 | 3 | 5 |
| **Global** | 1 | 1 | 4 | 3 | 5 |
| *Insertion*: | | | | | |
| Franka Lab 1 | 1 | 1 | 3 | 3 | 5 |
| Franka Lab 2 | 1 | 2 | 3 | 3 | 5 |
| **Global** | 1 | 1 | 3 | 3 | 5 |
| *Zipping*: | | | | | |
| Franka Lab 1 | 1 | 3 | 2 | 4 | 5 |
| Franka Lab 2 | 1 | 1 | 1 | 1 | 5 |
| **Global** | 1 | 1 | 4 | 1 | 5 |

Table 1: Rankings of five baseline approaches on four different tasks. We present local rankings in each lab and also compile data across experiments to compute global rankings.

lab level, and how broadly do these conclusions generalize? (3) And finally, can findings from multiple labs combine into a single useful global ranking?

## 6.1 Metric Level Analysis

RB2's full baseline suite was evaluated on all 4 tasks in 2 distinct lab spaces – **Franka Lab 1** and **Franka Lab 2**. These spaces featured 2 robots with the same hardware (Franka Pandas) in a tabletop setting. However, the robots were set in different lab spaces, used different camera setups, and were built on different platforms (pedestal v.s. custom tabletop). This realistic environment diversity was required to simulate lab spaces from different groups. The results are shown in Figure 3. The task metrics were normalized to $[0, 1]$ where a score of 1 signifies maximum possible performance. Furthermore, performance metrics for train and test objects were reported separately in order to enable further analysis. A cursory analysis reveals several comforting trends. For one, performance on test objects is consistently lower than for training objects, which strongly aligns with our expectations for neural network training. Furthermore, almost every baseline reports satisfactory performance on at least 1 task (e.g. zipping).

It is notable that one baseline - MOReL (Offline RL) [24] - significantly under-performs in every task. One possible explanation is that offline RL requires far more diverse "negative" data (i.e. low/zero reward state transitions) to train in a stable manner. To verify this, we collect a noise augmented dataset for the pouring task in Franka Lab 1. Specifically, the human expert demonstrations are played back with added noise and appended to the dataset using a similar procedure as in [8]. Furthermore, purely random trajectories are executed on the robot. Rewards from these trajectories are imputed using human labelling with the same method as before. While testing offline RL in this setting results in no boost in performance, qualitatively the trajectories seem more natural and tend the models tend to output less unstable behavior. An exception to this was pouring, where MOReL with no random noise would go close to the desired cup location but not pour. On the other hand, MOReL with random noise would pour to a similar location every time. This suggests that offline RL methods require significant amounts of data and have a long way to go before being competitive on RB2. We leave this as a challenge for future work.

**Findings:** Even before computing rankings, the data from this study supports a number of conclusions. For one, the zipping task is the easiest one in RB2 (though still not fully solved), while the insertion task is the hardest. Pouring and scooping are "medium" difficulty. Surprisingly, the simplest method, BC-Open, frequently outperforms more complicated methods like BC-LSTM, which has a sequential model, and MOReL, which requires reward signals. This serves as an important reminder that SOTA methods are not always complicated, and that many seemingly simple problems in robotics (e.g. learning closed loop controllers from expert data) remain stubbornly unsolved.

## 6.2 Method Rankings

While data contains several fascinating insights that holds across different labs, it is important to note that raw performance numbers do **not** in fact generalize across labs. This is an expected flaw that we seek to resolve using rankings. Specifically, we rank each method using a test success metric, for each lab in what we call *local rankings*. Due to the nature of hardware experiments and the number of test objects, we expect small deviations between these rankings. We combine the evaluations from all labs into *global rankings* and provide tests for statistical significance of these comparisons. All of our rankings are presented in Table 1.

**Local Ranking:** We begin by calculating rankings on a local level, by separately considering the data available to each lab. These rankings do reveal inconsistencies between labs. For example, the pouring evaluations from Franka Lab 1 suggests that NDP is significantly better than BC-Closed and BC-LSTM. However, in Franka Lab 2 the NDP model is better than the other methods, but not significantly so. Fortunately, certain trends do stably hold across all settings. For example, BC-Open always ranks higher than all other tasks consistently and emerges as true SOTA on these tasks. This trend suggests that local rankings are in general reliable and can be used for reporting results in future publications. However, to handle noise and ensure emergence of true SOTA, we compute global rankings as described below.

**Global Ranking:** Of course, there is always noise in local experiments that is impossible for small academic research labs to resolve. RB2's central proposition is to provide infrastructure for the field as a whole to reach consensus. This proposal is far more powerful than focusing on local research, since it democratizes access to knowledge in robotics and helps in emergence of SOTA via crowdsourcing to multiple research labs. The current global rankings and a mechanism to provide your own data for rankings is described on the website.

# 7 Conclusion

In this paper, we present a new robot benchmark RB2, alongside a new benchmarking philosophy. We argue that performance rankings between proposed algorithms and existing SOTA baselines are meaningful, so long as experiments were performed under the same setup. Therefore, RB2 is composed set of four benchmark tasks (pouring, scooping, zipping and insertion), *alongside* five SOTA baseline approaches (BC-Open, NDP, BC-Closed, BC-LSTM, MOReL). Furthermore, we show that evaluation data from multiple labs can be combined to create a singular global ranking. This allows users to both intelligently pick useful baselines using our freely available global rankings and implementations, and then demonstrate statistically significant improvements in their local labs. RB2 will evolve over time, as local advancements are contributed back to the repository. Indeed, we are already planning to replicate these experiments in more lab spaces, while using other robots. However, RB2 already reveals several notable findings: e.g. closed-loop learning approaches often struggle in the real world vs simple learned open-loop baselines; and simple behavior cloning outperforms much more complicated methods (like Offline RL). We sincerely hope robotics community uses this benchmark to democratize robotics and scale up the pace of research progress.

# 8 Acknowledgements

First, we'd like to acknowledge our collaborators at CMU, FAIR, NYU, WPI, and UIUC who helped make our final submission stronger. In particular, we'd like to thank Aditya Prakash, Kenneth Marino, and Zisu Dong. Finally, Lerrel Pinto was supported by ONR award numbers N00014-21-1-2404 and N00014-21-1-2758, David Held was supported by LG Electronics and the NSF CAREER Award (IIS-2046491), and Saurabh Gupta was partially supported by NASA Grant 80NSSC21K1030.

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
