# OpenReview forum: "RB2: Robotic Manipulation Benchmarking with a Twist"
_NeurIPS.cc/2021/Track/Datasets_and_Benchmarks/Round2 — NeurIPS 2021 Datasets and Benchmarks Track (Round 2)_

### Official Review · Reviewer_m7iM · 2021-09-19
**Review for RB2: Robotics Benchmarking with a Twist**

**Rating:** 7
**Confidence:** 3
**Correctness:** Everything appears to be correct (to …

**Strengths:**

1. The paper considers real-world uncertainties when constructing benchmarking results
2. The framework is expandable by adding more task and lab settings and can be adopted by other research groups
3. The benchmarking process is comprehensively explained

**Weaknesses:**

I would like to ask the authors few clarification questions:
1. In the proposed system of global ranking between different research groups with different robotic setups, would it make it easy for someone or a group of people to game the system by evaluating their models in an easier setting and influencing global ranking ?
2. How easy would it be to expand this system to include more tasks and benchmarking models? More specifically are you proposing to create and maintain this system and accept submissions or is it more of a guideline for others to build their ranking systems.


**Additional Feedback:**

I don't have any additional feedback

**Clarity:**

The benchmarking process is comprehensively explained. The results on all 4 tasks with all the models are clearly shown in the graphs and table.


**Documentation:**

I tried going to this website "https://www.rb2.info/" mentioned in paper. But I think it is incomplete with the link to the code broken. The website has some documentation to explain 4 tasks.

**Ethics:**

I don't believe there are any ethical concerns as the task presented are very basic

**Relation To Prior Work:**

The paper doesn't claim to be first of a kind benchmark but discusses how it differs from previous contributions


**Summary And Contributions:**

This paper presents a robot benchmark that includes 4 tasks (scooping, pouring, zipping, and insertion) that can be performed by a robot under different physical settings. The paper focuses on defining the benchmarking process for robotics that accounts for real-world uncertainties. The presented tasks are evaluated on existing SOTA models. The paper concludes by showing some SOTA models are indeed robust to changing conditions. Their presented methodology can also be adopted by other researchers for their environments or tasks, and models.

---

> ### Author Response · Authors · 2021-09-27
> **Response to m7iM**
>
> Thanks for your feedback. We've attempted to answer your questions/concerns below, and have provided some additional context above in the common response post.
>
>
> >In the proposed system of global ranking between different research groups with different robotic setups, would it make it easy for someone or a group of people to game the system by evaluating their models in an easier setting and influencing global ranking ?
>
> Note that the global rankings are self correcting. As more groups try to replicate a “false positive” result, the new data will show that the original experiment was an outlier. If the error is significant/malicious, the wrong information can be deleted from the central database. Of course, this system will break down if most researchers in the field are malicious, but no real world benchmark could realistically survive such an environment.
>
> >How easy would it be to expand this system to include more tasks and benchmarking models? More specifically are you proposing to create and maintain this system and accept submissions or is it more of a guideline for others to build their ranking systems.
>
> We plan to accept adding new baselines into the system, by allowing researchers to upload implementations into our model zoo. Indeed, this will be a core requirement for submitting a method to our global ranking system.
>
> In addition, we would be excited to work with others on adding new tasks to the benchmark. In theory, it should be a straightforward replication of the work we did in this paper but for the new tasks. That being said, no specific system for adding tasks is planned at this time until we see saturation in the current performance metrics.

---

### Official Review · Reviewer_3bqf · 2021-09-20
**A robotic manipulation benchmark to that works over with heterogeneous hardware and environments**

**Rating:** 5
**Confidence:** 4

**Strengths:**

Evaluating the performance of robots is challenging due to the fact that they operate in the real world, in environments that cannot be fully controlled and that no two robots are identical, even those of the same model. This paper presents a method to mitigate these challenges by requiring the use of a set of baselines and then using the relative local rankings of a proposed method versus the baseline as the evaluation metric.

**Weaknesses:**

To get accurate results requires significant buy in by the community. Adding in a different set of items might be an issue. Using the same objects as YCB objects might mitigate this.
Limiting the tasks to manipulation with objects already in the robot’s hand might be too restrictive.

**Additional Feedback:**

It would be interesting to see if there was a way to integrate previous benchmarks if they include a local comparison to a baseline.


**Clarity:**

The paper was clear and easy to follow.


**Correctness:**

The training/test splits should be universal, comparing policies trained on scooping nuts and tested on coffee might not be similar enough to one trained on coffee and nuts and tested on nuts.

**Documentation:**

The website and paper did provide enough information to get started, and the code to produce the baselines should be on the linked website, but it was not finished at the time of the review.


**Ethics:**

There are not any ethical concerns.


**Relation To Prior Work:**

The paper discusses other robotics benchmarks, which are focused on specific tasks or hardware, while this work proposes a general approach across diverse hardware and multiple tasks.

**Summary And Contributions:**

The paper presents a method of allowing for benchmarks across a variety of hardware and environments. In the proposed RB2 benchmark new algorithms are tested alongside current baselines over four manipulation tasks. These relative local results are then passed to a centralized system where multiple local rankings are aggregated into  global rankings.

---

> ### Author Response · Authors · 2021-09-27
> **Response to 3bqf**
>
> Thanks for your feedback. We've attempted to answer your questions/concerns below, and have provided some additional context above in the common response post.
>
> > To get accurate results requires significant buy in by the community. Adding in a different set of items might be an issue. Using the same objects as YCB objects might mitigate this. Limiting the tasks to manipulation with objects already in the robot’s hand might be too restrictive.
>
> Of course, making any benchmark successful requires significant buy in from the community. We hope RB2’s advantages (e.g. global ranking system, reproducible tasks, open source implementations, etc.) will help convince other researchers to adopt this.
>
> That being said, we agree that using existing objects is always better. While YCB unfortunately does not contain objects necessary for all our tasks, we are open to replacing some of our test/train objects with those from YCB.
>
>
> > The training/test splits should be universal, comparing policies trained on scooping nuts and tested on coffee might not be similar enough to one trained on coffee and nuts and tested on nuts.
>
> The train/test sets are listed on our website ([rb2.info](https://rb2.info)) and are easy to acquire online or at local supermarkets. This should allow other researchers to replicate our object sets, while making reasonable substitutions where necessary.
>
> Furthermore, our ranking scheme was designed explicitly to deal with the issue you highlighted. While raw numbers for policies tested on nuts may be different from those tested on coffee, the relative differences should remain constant so long as all models are locally trained/tested in a consistent manner. This should allow us to make fair comparisons even when the exact training/testing objects do not match.
>
> ========================================= EDITED on 9/29  ============================================
> >It would be interesting to see if there was a way to integrate previous benchmarks if they include a local comparison to a baseline.
>
> We believe our two step benchmarking strategy is effective and therefore should be able to incorporate results and methods from previous benchmarks into RB2. We envision using previous benchmark rankings as another local ranking. If one of the labs is able to create a local ranking with previous benchmarks, then combining it into the global ranking should be effective. In the future, as the advisory committee evolves, we do envision working closely with previous benchmarks to help bootstrap new tasks into our benchmarks.

---

> > ### Author Response · Authors · 2021-09-29
> > **Response Follow-Up**
> >
> > We hope we have fully answered the concerns you raised earlier, regarding the website and the benchmark's object set. Please let us know if your comments on these points are addressed fully, or if you have more questions.

---

### Official Review · Reviewer_kAx9 · 2021-09-20
**Intersting and useful technique for robotic benchmarking**

**Rating:** 7
**Confidence:** 3
**Clarity:** The paper is written clearly.

**Strengths:**

1. Interesting idea: This is a simple but interesting idea to ensure that new models and algorithms that are claimed to be state-of-the-art actually offer benefits over other pre-existing techniques in settings that follow some standardized experimentation and task instantiation protocols. That helps in providing a level playing field for most researchers and reduces the chance of optimizing the evaluation setting(s) to suit one's own model/algorithm.

2. Model zoo: the paper tests 5 strong baselines in 2 different settings with very similar experimentation protocols and task instantiation parameters and find trends about the relative strengths of the algorithms. This not only reveals which algorithms actually work better but also allows a new researcher in the area to efficiently run experiments because they know which model is the one to beat.


**Weaknesses:**

1. L155, 'Note that individual ... their results to a central repository': If a newly proposed model isn't uploaded to the model zoo, how would the global ranking be generated? As far as I understand, each new model should be tested across some standard lab environments and the results from these trials should be aggregated to finally rank all the models.

2. Global ranking algorithm: The paper doesn't clearly mention how the rankings from the 2 labs are aggregated to generate the final global ranking (table 1).


Edit on 9/29:
My concerns have been adequately addressed. I vote for accepting the submission. I am changing my rating for the paper accordingly.

**Additional Feedback:**

Minor typos:
1. L309, 'Thus, Our final': 'Thus, our final'?

2. L255, 'and tend the models tend': 'and the models tend'?

**Correctness:**

The claims made in the paper and the supporting experiments/evaluation methods look correct to me.

**Documentation:**

I couldn't access the benchmark website. I have tried with both Chrome and Firefox from both Ubuntu and Windows but both give me a warning about accessing a site that's not secure. It could have something to do with the security certificate. Maybe you would want to fix it.

**Ethics:**

I couldn't spot any ethical issues.

**Relation To Prior Work:**

The work compares with prior work and tries to identify issues with existing benchmarks.

**Summary And Contributions:**

This work proposes a way to benchmark robotic algorithms so that the findings with every new algorithm are more or less consistent across different evaluation settings, and add real value to robotics research. The contributions of this work are two-fold:

1. It introduces a method to evaluate a proposed robotic algorithm in settings that rigidly enforce experimentation protocols but could differ in system/intrinsic parameters. Following the evaluations, the models in consideration are assigned a global rank by combining the local ranks across different settings. This helps in testing the generalization capability of an algorithm and identify models that are better than others in the majority of the settings.

2. It also introduces 4 tasks of varying difficulty that could be replicated across multiple laboratories with variations only in the system/intrinsic parameters of the environment, agent, and the object(s) involved, and 5 state-of-the-art baselines on these tasks that have been tested in 2 different lab environments. This repository of baselines and their results on the benchmark could offer useful trends and insights needed in designing even stronger models, and also allow for making systematic progress in the tasks.

---

> ### Author Response · Authors · 2021-09-27
> **Response to kAx9**
>
> Thank you for your feedback. We've attempted to answer your questions/concerns below, and have provided some additional context above in the common response post.
>
> > L155, 'Note that individual ... their results to a central repository': If a newly proposed model isn't uploaded to the model zoo, how would the global ranking be generated? As far as I understand, each new model should be tested across some standard lab environments and the results from these trials should be aggregated to finally rank all the models.
>
> The global ranking is created by pooling local rankings, so calculating a preliminary ranking for a new method should be easy. The local rankings and results will be submitted to the central database of all other methods. Once these are submitted, we will process them to 1) filter outliers and check if the submissions match the standard. We will then re-compute the rankings using the Plackett-Luce method (See the common response to all reviewers).
>
> >Global ranking algorithm: The paper doesn't clearly mention how the rankings from the 2 labs are aggregated to generate the final global ranking (table 1).
>
> Please see the common response to all reviewers for details on Plackett-Luce method for global rankings from several local rankings. We will add a more thorough discussion of this in the paper.
>
> >I couldn't access the benchmark website. I have tried with both Chrome and Firefox from both Ubuntu and Windows but both give me a warning about accessing a site that's not secure. It could have something to do with the security certificate. Maybe you would want to fix it.
>
> Thank you for pointing this out. You should be able to access the website by typing [rb2.info](https://rb2.info) (no www etc) into your browser. We are actively looking into what is wrong with the certificate.
>
> >Minor typos:
> >L309, 'Thus, Our final': 'Thus, our final'?
> >L255, 'and tend the models tend': 'and the models tend'?
>
> Thanks for noticing these! We will be sure to fix them in the final version.

---

> > ### Author Response · Authors · 2021-09-29
> > **Response Follow-up**
> >
> > We hope we have fully answered the concerns you raised earlier, regarding the website and the benchmark's global ranking mechanism. Please let us know if your comments on these points are addressed fully, or if you have more questions.

---

> > > ### Comment · Reviewer_kAx9 · 2021-09-29
> > > **Response to rebuttal**
> > >
> > > Yes, my concerns have been adequately addressed. I vote for accepting the submission. I am changing my rating for the paper accordingly.

---

### Official Review · Reviewer_LkWv · 2021-09-21
**A step towards standardizing manipulation experiments but questions remain**

**Rating:** 7
**Confidence:** 3

**Strengths:**

The paper addresses a serious problem in robotics research that is still largely unsolved and hinders progress in the field. Although the proposed framework only addressed the task of manipulation, it contributes ideas and findings to address these issues.

**Weaknesses:**

The implementation and practicality of such an implementation remain unclear.

Scalability and missing experiments:
a) From the current description (line 143-144), the researcher that is testing their new algorithm must locally reproduce the existing baselines. As the number of baseline grows this becomes more and more tedious and impractical. How is the case of missing local baselines handled?

b) As the number of baselines grows, it becomes impractical that N labs maintain all M baselines. Some labs might only be able to implement a subset of the baselines. How is this handled in the current rankink scheme?

Using this benchmark:
Let's assume I am a researcher who would like to publish a new method and compare against the baselines of R2B. Let's also assume that I implemented all the existing baselines of the benchmark locally. No global ranking of my method exists yet, so I can only report my local ranking. How does this local ranking help the reviewer of my research paper to assess the proposed method? To my understanding, the proposed benchmark only provides additional information through its global ranking. As such, my method will only be compared against if all the labs contributing to the benchmark also implement my method.

Malicious intent:
It cannot be ruled out that research groups deliberately submit weak implementations of competing approaches/groups. How can this be addressed?

Experiments:
a) Section 5.4 introduces a vision front-end that is very simplistic together with the following sentence: "We want to focus RB2 on measuring manipulation performance". However, vision is used in closed-loop in many manipulation approaches and thus an integral part of solving the problem. What is the reasoning behind trying to exclude the vision pipeline from the benchmark? Are the researchers contributing to RB2 tied to using the simple model proposed in this paper? If not, please clarify this statement.

b) It would have been insightful to add a third research group that only partially contributes to the benchmark. This way it would be clear how this case is handled.

**Additional Feedback:**

The title suggests a generic robotics benchmark while the paper is specifically about manipulation. This must be adapted to avoid confusion.

My rating is based on the fact that there are several open questions about implementation, utility, and documentation. I am open to raising the rating if my concerns are properly addressed. Overall, the proposed approach is a step in the right direction.

**Clarity:**

See weakness section. Missing clarity in how the benchmark implements corner cases of missing experiments and how the benchmark helps in the review process.

**Correctness:**

The evaluation methodology, as described in the paper, is constructed in a sound way. The experiments seem appropriate.

**Documentation:**

The documentation is insufficient. As of this review, the website points to a website that does not contain information about code or documentation. The management of the project in terms of availability and maintenance is not clear from either the paper or the website.

**Ethics:**

No ethical concerns

**Relation To Prior Work:**

Yes, the difference to prior work is clearly discussed

**Summary And Contributions:**

The paper proposes a robotic benchmark with focus on manipulation tasks. It addresses the longstanding problem of reproducibility of real-world robotic experiments. The proposed contribution is a framework that relies on a set of baselines to be implemented by multiple research labs to create a global ranking. This partially addresses the problem of variability in the setup that leads to different outcomes in different research labs.

---

> ### Author Response · Authors · 2021-09-27
> **Response to LkWv. [1/2]**
>
> We thank you for the feedback and for pointing out several missing implementation details. First we would highlight that we have explained the dynamics of how benchmark and code repository evolves with a fictional example in the common response. We will now try to answer your questions point by point.
>
> > Scalability:  As the number of baseline grows this becomes more and more tedious and impractical. How is the case of missing local baselines handled?
>
> This is a great point and definitely something we missed discussing in the paper. Since the goal of the benchmark is in discovering and creating new state of the art results, we believe the local labs will only need to compare their work against the top (e.g. best 3) methods currently in the benchmark. This will ensure that new advancements can be fairly judged, while keeping the amount of work required for the benchmark constant. Our global ranking function handles missing ranks of lower baselines well (see partial rankings explanation in common response to all reviewers).
>
> >As the number of baselines grows, it becomes impractical that N labs maintain all M baselines. Some labs might only be able to implement a subset of the baselines. How is this handled in the current ranking scheme?
>
> Yes, this is handled by the partial local ranking to global ranking system, which is described in the common response above.
>
> > Using this benchmark: Let's assume I am a researcher who would like to publish a new method and compare against the baselines of R2B. Let's also assume that I implemented all the existing baselines of the benchmark locally. No global ranking of my method exists yet, so I can only report my local ranking. How does this local ranking help the reviewer of my research paper to assess the proposed method? To my understanding, the proposed benchmark only provides additional information through its global ranking. As such, my method will only be compared against if all the labs contributing to the benchmark also implement my method.
>
> The first step for researchers is to compute local rankings v.s the top baselines in their own labs.While they won’t have the same weight as a global ranking, these local rankings will still offer a more scientific way to judge new methods against existing work. It is far preferable to the status quo in robotics, where even tasks are not standardized across papers.
>
> But how would global rankings for the method appear?  For any approach to appear on our global leaderboard, they will have to provide a description of their approach and also the code compatible with the evaluation system. Note our benchmark repository consists of all baseline code already. Once the code is uploaded, some other laboratory will use this code to compare their results and hence provide data points for global ranking to be computed by the field over time.
>
>
> > Malicious intent: It cannot be ruled out that research groups deliberately submit weak implementations of competing approaches/groups. How can this be addressed?
>
> We believe that the confusion is that a research group is expected to submit the implementation of their own algorithm, while only submitting rankings of the baselines (v.s their new method). We believe it is unlikely that researchers will maliciously submit bad numbers to our global ranking. However, in that case other research labs would try to reproduce these results and baselines as part of their own experiments. This reproduction would yield honest results, thus allowing the rankings to self correct. Furthermore, our server supports rejecting erroneous/fraudulent submissions after the fact, which will allow us to mitigate damage once detected. If anything, RB2’s transparent and collaborative nature would make fraud detection easier than the status quo, where crucial details are often ignored/kept under wraps.

---

> > ### Author Response · Authors · 2021-09-27
> > **Response to LkWv [2/2]**
> >
> > > Experiments: a) Section 5.4 introduces a vision front-end that is very simplistic together with the following sentence: "We want to focus RB2 on measuring manipulation performance". However, vision is used in closed-loop in many manipulation approaches and thus an integral part of solving the problem. What is the reasoning behind trying to exclude the vision pipeline from the benchmark? Are the researchers contributing to RB2 tied to using the simple model proposed in this paper? If not, please clarify this statement.
> >
> > First, thank you for pointing out this error in framing. We do not believe the vision pipeline is tied to this setup. In fact, we believe this benchmark could serve as a way to compare different existing vision pipelines and see which works the best. However, we wanted to provide a reasonable initialization point that would allow control/robotics researchers to quickly get started on vision based manipulation tasks, without much worry. Of course, not everyone is constrained to using this setup, and more investigation on the vision module is encouraged! We will make this clearer in the writing.
> >
> > > b) It would have been insightful to add a third research group that only partially contributes to the benchmark. This way it would be clear how this case is handled.
> >
> > Note that it is easy to accept experiments that only test on a single task (e.g. just pouring). Furthermore, we plan to support groups that only compare against a few (e.g. the best 3) methods in our global rankings (within a task). The specific mechanism enabling this (Plackett-Luce method) is discussed in the common response, and we are planning to perform more verification experiments (at UIUC/FAIR) to test this case explicitly.

---

> > > ### Comment · Reviewer_LkWv · 2021-09-29
> > > **Response from Reviewer LkWv**
> > >
> > > Thanks for the clarifications.
> > >
> > > Two points are missing from your response:
> > > 1) Your paper title, as well as the abstract, does not mention that this paper is about manipulation. You cannot claim generality for all of the robotics research because this approach is designed for manipulation. Will you clarify in either the title and/or abstract the scope of this work?
> > > 2) Who is maintaining this project? Is it a single person or a lab or multiple labs?
> > >
> > > Thanks for uploading the code. I see that it is hosted by a personal account and not an organization. The title, as well as the number of authors, suggest that this is a larger project that is planned to be maintained over some years by multiple labs. I suggest creating a GitHub organization for such a project. This last comment/paragraph is not to be interpreted as a scientific review that affects the rating but as an outside perspective about the credibility of the project.

---

> > > > ### Author Response · Authors · 2021-09-29
> > > > **Response to LkWv**
> > > >
> > > > First, we hope we have fully answered the concerns you raised earlier, regarding how to handle partial ordering and benchmark evolution. Please let us know if your comments on these points are addressed fully, or if you have more questions.
> > > >
> > > > Furthermore, we apologize for missing these questions before and will answer them now:
> > > >
> > > > > Your paper title, as well as the abstract, does not mention that this paper is about manipulation. You cannot claim generality for all of the robotics research because this approach is designed for manipulation. Will you clarify in either the title and/or abstract the scope of this work?
> > > >
> > > > Thank you for this catch. We are working on revisions to the paper at the moment, and will be sure to specify manipulation more clearly in both the title and abstract.
> > > >
> > > > > Who is maintaining this project? Is it a single person or a lab or multiple labs?
> > > >
> > > > Our plan is for the benchmark's day to day operations to be maintained by a group for a single term (length 2-3 years). Specifically, CMU will take the this role first. We will automate as many parts of the benchmark as possible to reduce load on this group. In addition, we will have an advisory committee of 5 people to oversee RB2's long-term trajectory. At end of each "maintenance term," the advisory committee will work together to find new set of people/group to take over.
> > > >
> > > > > I suggest creating a GitHub organization for such a project.
> > > >
> > > > Finally, thank you for the suggestions on the code base. We will definitely make a more official github org/model zoo, and are planning to make the presentation (in general) more professional.

---

> > > > > ### Comment · Reviewer_LkWv · 2021-09-29
> > > > > **Final Comment**
> > > > >
> > > > > Thank you for answering all of my questions.
> > > > >
> > > > > Assuming the authors take all the comments and suggestions into account, I vote for accepting this submission.

---

### Author Response · Authors · 2021-09-27
**Common Response: Clarifying Questions Regarding Global Ranking and Website**

First, we want to thank the reviewers for all their feedback. Before we answer individual queries, we would like to take a moment to contextualize our work. Unlike other ML/AI fields such as computer vision, NLP, or even Reinforcement Learning, there is a big void when it comes to benchmarking in robot learning. Our work is an attempt to fill that void. Does our benchmark have shortcomings? Is it impractical? Can it be gamed? Perhaps so, but we believe doing nothing and not having a benchmark is much worse. That being said, we are grateful for all the feedback since it should make the benchmark even stronger. We hope that our response should be able to handle most (if not all) of the concerns raised by the reviewers. We will make all the necessary changes and upload before the deadline.

Based on the reviewer’s questions, we believe there is some confusion about how our benchmark and code repository would evolve - particularly when users attempt to contribute to global rankings with partial evaluations. We will update the paper with a more thorough discussion of these points. For now, we will give further details and show a fictional case study, in the next section of this response.

Let us assume at a given time, our code repository has 5 baseline algorithms implementations ($B_1$-$B_5$). Currently the global rankings show: $B_3 > B_2 > B_5 > B_4 > B_1$. Another research group proposes a new algorithm $A_1$ after our baselines are pushed. To demonstrate their algorithm is better they download the code of the top-3 baselines ($B_3$, $B_2$ and $B_5$) and compare their proposed method ($A_1$) to the baselines. This comparison is published using local rankings. In order to aggregate these results into the global ranking, the authors submit their paper, results and code to our repository. To build a global ranking, we take local ranking orderings ($A_1, B_3, B_2,$ etc). These rankings can be partial (only have $m$ out of $N$ possible methods) or have possible ties in the rankings. We employ the Plackett-Luce method which firstly builds a distribution over the set of rankings and then performs MLE to find the global ranking. More details about the Plackett-Luce method can be found in the textbook by Marden, 1995 [1]. Once the code is submitted the top-3 suggested baselines now include $A_1$ (Note the global rankings still does not include $A_1$. To appear in global rankings, $A_1$ needs to be run by another independent research group). If another group proposes method $A_2$, they will have to download code for $A_1, B_3$ and $B_2$ (the top 3 current baselines) and compare against these. Similarly to $A_1$, they will run experiments, collect local rankings and results which they will submit to our repository alongside their implementation. Once $A_2$ uploads their local rankings, our global rankings get updated to reflect $A_1$ as the new leader since their results have been confirmed by another lab.

Finally, we would like to apologize for the website, which can be accessed at [rb2.info](https://rb2.info). Our most up-to-date revision did not get pushed onto the server. We have now updated the website with better documentation, video tutorials for each task, and a link to our baseline github repository. Our benchmark is currently being independently replicated at multiple other sites including FAIR and UIUC, and we believe that feedback from this process will yield further improvements.


[1]. Marden, J. (1995). Analyzing and modeling rank data. Chapman and Hall.

---

### Decision · Program_Chairs · 2021-10-09

**Decision:**

Accept

**Comment:**

This paper presents a method of benchmarking robotic manipulation tasks across a variety of hardware and environments. The core idea is that multiple different groups implement a set of baselines, and the local performances on each baseline are combined into a global ranking against which work can be compared. The paper introduces four baseline tasks, which are tested in two different research settings.

The approach proposed in the paper addresses a real existing problem in robotics, where different platforms and environments tend to make comparison of research results difficult. The approach is clearly and thoroughly explained. As conceived, the ranking approach is amenable to being extended with additional baselines and is constructed to have additional research groups join. However, some concerns remain. The problem explicitly omits the vision and grasping components of the manipulation pipeline, which may not be appropriate in all cases. More seriously, when a new method/model is proposed, it must be implemented by multiple groups in order to receive a global ranking, and it's not clear how the purely local ranking should be considered; both for this reason and in general, the proposed benchmark is of most use when there is significant buy-in from multiple research groups, which may be difficult to bootstrap. Reviewers also have concerns about the potential for bad actors to affect the system by maliciously manipulating their claimed local results; however, these concerns are consistent with the current state of the art in robotics. Authors have engaged constructively with reviewers, and are encouraged to thoroughly incorporate improvements from these discussions in subsequent versions of the paper.